# Why Employees Contribute to Pro-Environmental Behaviour: The Role of Pluralistic Ignorance in Chinese Society

**Hao-Fan Chumg [1], Jia-Wen Shi [2] and Kai-Jun Sun [3],***

[1] School of Management Engineering, Nanjing University of Information Science & Technology, Nanjing 210044, Jiangsu, China; h.chumg@nuist.edu.cn

[2] School of Liberal Arts and Sciences at University of Greenwich, London SE10 9LS, UK; shijiawenn@gmail.com

[3] Department of Computing and Software Engineering, Huaiyin Institute of Technology, Huaian 223003, Jiangsu, China

* Correspondence: sunkaijun9@gmail.com; +86-198-250-55-165

**Abstract:** In light of the importance of sustainable development, this study aims to deepen and extend our understanding of employees' pro-environmental behaviour in the workplace in a Chinese context. Drawing on the complex phenomenon of social norms theory concerning misperceptions (i.e., pluralistic ignorance) and supervisor–subordinate *guanxi* (which is a Chinese term signifying human connection), we present a novel model in which employees' pro-environmental behaviour is the result of multiple social and individual psychological factors. Through the integration of previous literature from the fields of the psychology of individuals, social psychology, and environmental psychology, the major assumption is that the pro-environmental behaviour of employees is affected by their level of pluralistic ignorance, environmental concern, and subjective norms; these, in turn, are influenced by supervisor–subordinate *guanxi* and social identity in the collective spirit of Chinese society. Data, which were analysed empirically, were gathered from 548 Chinese employees from the Jiangsu province of China. This study consequently reveals the subtle interplay among employees' pluralistic ignorance, supervisor–subordinate *guanxi*, social identity, subjective norms, environmental concern, and their pro-environmental behaviour, while the deeper analysis offers considerable support for environmental management research and practice.

**Keywords:** pro-environmental behaviour; pluralistic ignorance; social norms; supervisor–subordinate *guanxi*; social identity; subjective norm

## 1. Introduction

Due to the burgeoning of environmental awareness in China and the ever-accelerating development of its economy, the Chinese government, environmental protection organisations, and citizens are now paying more attention to the ecological deterioration of the environment in order to improve sustainability [1,2]. However, according to the latest environmental performance index (EPI) report (2018), China was ranked 120th (environmental performance index), 167th (environmental health), and 177th (air quality) out of 180 recorded countries (Environmental Performance Index, 2018). It is undeniable that, even though the Chinese government makes considerable efforts regarding environmental enhancement, most of these efforts seem to ultimately fail. The continuous pollution and environmental degradation in China require a closer look to be taken at factors that influence Chinese citizens' environmental attitudes and behaviour [3]. This is mainly because Chinese people tend to hold an anthropocentric perspective, valuing more what the environment can do for them and

so choosing economic growth rather than considering environmental protection [4]. Sun [5] further states that most Chinese people, because they generally lack a sense of personal responsibility, tend to think that the government, rather than they themselves, has a duty to protect the environment. Undoubtedly, while human behaviour has a significant influence on the capability of the earth to sustain and cultivate all life forms [6], most organisations and institutions ignore the support of employees' pro-environmental behaviour [7].

On that account, to solve the deteriorating environmental issues, the Chinese government has eventually utilised the evaluation system, regarded as a powerful tool for estimating officials' promotion, asking officials to assess organisations' managers in terms of improving the protection of the environment [8]. Also, if organisations fail to achieve the goals of environmental performance supervised by the relevant officials, their promotion prospects would be influenced [8]. By contrast, an increasing number of organisations are paying more attention to corporate environmental behaviour in order to maintain good *guanxi* and also to obtain the financial capital offered by Chinese government officials. In fact, the environmental behaviour of Chinese firms has been significantly affected by their employees, the government, and relevant industrial associations [2,9]. Meanwhile, employees spend about one-third of their day in the workplace and therefore their daily pro-environmental behaviour is very helpful in reducing the negative effect of workplace activities on the company's environment [10]. Employees' pro-environmental behaviour includes recycling, waste management, or any other ecological behaviour that has an environmentally friendly effect on the environment [11]. Employees are therefore encouraged more than ever by Chinese firms and the government to improve their environmental performance. Prior research has shown that pressure resulting from norms might explain certain discrepancies in the pro-environmental behaviour of Chinese people [12]. However, it seems more difficult to transform the attitudes of individuals towards environmental protection into real behaviour since an individual's pro-environmental behaviour depends heavily on contingent factors and the societal context [13].

Despite a gradual increase in the academic literature exploring the cause-and-effect of pro-environmental behaviour, little is known about the extent to which the enhancement of employees' pro-environmental behaviour can be ascribed to misperceptions of social norms in a Chinese social context. In this study, we argue that multiple social norms of employees, especially pluralistic ignorance, should be further explored for the purpose of examining the sociological mechanisms of employees' responses to pro-environmental behaviours. Furthermore, in this study, social norms theory has been extended to offer a comprehensive understanding of the underlying mechanisms influencing the response of employees to pro-environmental behaviour. The main objectives of this research are, in particular, to:

(1) Analyse the correlation between social psychological factors (i.e., supervisor–subordinate *guanxi* and social identity) and employees' pluralistic ignorance;

(2) Analyse the influence of employees' pluralistic ignorance on their subjective norms;

(3) Analyse the role of employees' pluralistic ignorance associated with their pro-environmental behaviour;

(4) Analyse the mediating effect of employees' pluralistic ignorance on their supervisor–subordinate *guanxi*, social identity, and pro-environmental behaviour within an organisation in the context of Chinese contemporary culture.

## 2. Conceptual Background and the Development of Hypotheses

*2.1. The Concept of Misperceived Social Norms Theory Applied in This Research*

Social norms refer to the expectation of appropriate behaviour from an individual that occurs in a specific group context. They have been conceptually regarded as "rules of conduct", which are maintained in part by pros and cons. A social norm can be deemed to be a fundamental essence of human life, which consists of culture, language, social interaction, prejudice, and economic exchange [14]. Young [15] further describes social norms as unwritten rules that guide individuals' behaviour; they define what we expect of others and vice versa. Moreover, when compared with laws and codes, which are regarded as explicit, Farrow et al. [16] state that social norms are relatively inherent and invisible, placing them outside laws and other clearly defined social frameworks.

Prior research studies have consistently found the significant and pervasive impact of social norms on human behaviour [17]. Humans pay more and careful attention to observing others and imitate what they see; however, such learning by imitation is not sufficient to implicate social norms [14]. Social norms therefore are not pure imitations but rely on the expectation of others in terms of when imitation is appropriate and when it is inappropriate in a group. Academics have gradually drawn on the "misperceptions" derived from the concept of social norms to explain humans' social behaviour. These normative misperceptions result in false uniqueness, false consensus, and pluralistic ignorance [18]. The term "misperception" can be described as individuals' cognitive bias between actual behavioural intention and people's true views of the attitudes or behaviours of others. Hence, a misperception occurs when the benefits of attitudes and behaviours within a group are overestimated or underestimated. Individuals may misperceive their group in various ways that affect their behaviour. Meanwhile, each misconception works in a different way and can have different effects on behaviour [19]. False consensus occurs when a minority of people mistakenly consider that they are in the majority and others share their beliefs [20]. Pluralistic ignorance, by contrast, is a very common misperception that occurs when a majority of people may wrongly consider that they are in the minority although their attitudes or behaviours are more representative than they presume [20,21]. Finally, false uniqueness may occur when an individual considers that his/her attitude or behaviour is more particular than others [19,22].

With an acknowledgement of the significant influence of social norms theory on transforming the behaviour of individuals, this study extends and deepens the concept of misperceived social norms theory for the purpose of exploring employees' pro-environmental behaviour in the workplace in a Chinese context. In this research, it is claimed that misperceived social norms theory will significantly explain employees' responses to pro-environmental behaviours. It is also considered here that the interplay of overlaying multiple social norms helps to explain the acceptance and adoption of Chinese employees' behaviours and attitudes towards environmental protection and pro-environmental behaviour. Relevant literature concerning pluralistic ignorance is integrated in the following table, Table 1.

**Table 1.** Relevant literature integrated in this research concerning pluralistic ignorance.

| Authors | Context | Major Findings |
| --- | --- | --- |
| Nakashima and Flynn [20] | Social projection may improve willingness to participate in generalised exchanges. | False consensus not only makes individuals more inclined to engage in generalized exchanges, but also leads to more successful exchanges through a favourable process of self-selection. |
| Warner and Burchfield [23] | This study explores the influence of pluralistic ignorance on community values with regard to the possibility of informal social control. | Misunderstood values are significantly associated with the level of informal social control. |
| Boon, Watkins, and Sciban [24] | The authors tested the relationship between pluralistic ignorance and faith in terms of infidelity in dating relationships. | Individuals think that others are more likely to engage in unfaithful dating than they are. |
| Munsch, Ridgeway, and Williams [25] | The research experimentally tested the notion that workers view other workers who are more flexible less positively than they actually do. | The study found that flexibility bias stems partly from pluralistic ignorance, and when the majority of high-ranking employees work flexibly, the bias against workers of flexible hours (but not flexplace workers) decreases. |
| De Larios and Lang [26] | The authors examined the impact of pluralistic ignorance within the context of the virtual community. | Compared with players in the real world, pluralistic ignorance has a demonstrably lower presence in members of a virtual community who are more inclined to maintain consistency in their personal attitudes and public behaviour. |
| Brener et al. [27] | The study explored whether the hypothetical attitudes of colleagues, rather than their own, predicted health workers' behavioural intentions towards others who inject drugs. | Participants supported more harm reduction services for injecting drug users than they thought because of pluralistic ignorance. |
| Halbesleben et al. [28] | The study attempted to understand the impact of pluralistic ignorance on students' unethical behaviour. | Pluralistic ignorance plays an important role in affecting individuals who misinterpret unethical behaviour of other people. |
| Levine et al. [29] | The study explored why student-athletes perform poorly in academia. | Most student-athletes considered that they achieved very good academic performance; they also thought that their peers did not. |

*2.2. Environmental Concern and Employees' Pro-Environmental Behaviour*

Environmental concern can be deemed to signify an individual's general or global attitude towards green issues [30]. This is supported by Dunlap and Jones [31] who described environmental concern as to what extent human beings have the awareness of environmental problems and make great efforts to solve these issues. Luo and Deng [32] defined environmental concern as the combination of faith, affection, and behavioural intentions that an individual holds concerning environmental issues and activities. Hence, environmental concern can be refined to reflect an individual's basic attitudes, which consist of the cognitive and emotional evaluation of environmental problems, with antecedents of perspectives, knowledge, and beliefs concerning environmental matters [33,34]. Environmental concern can also be used interchangeably with the term "environmental attitude" while environmental concern has long been considered as an important explanation for the extent to which individuals engage in sustainable-oriented behaviour [32,33,35]. Therefore, this research adopts this concept in its investigation and reasonably assumes that environmental concern would improve individuals' behaviour in a more environmentally friendly way. This leads to hypothesis H1.

**Hypothesis (H1).** *Environmental concern positively affects employees' pro-environmental behaviour in an organisation.*

*2.3. Social Norms and Employees' Pro-Environmental Behaviour*

Subjective norms refer to social norms, which stem from the theory of reasoned action (TRA) [36]; they regulate behaviour that should be performed or not be performed by considering perceived social pressure [37]. Paek et al. [38] further describe subjective norms as an individual's belief that significant others affect whether or not the behaviour should be performed. Previous research studies have shown the significant impact of subjective norms on explaining individuals' pro-environmental intentions and purchasing behaviour [39–41]. As a result, an individual may be passive or reluctant to participate in environmentally friendly activities when he/she perceives some degree of social pressure [41]. Hence, this study reasonably assumes that subjective norms would improve individuals' pro-environmental behaviour in a more environmentally friendly way. Subsequently, we hypothesise:

**Hypothesis (H2).** *Social norms of employees positively affect their pro-environmental behaviour in an organisation.*

*2.4. Subjective Norms and Pluralistic Ignorance*

A greater level of pluralistic ignorance in individuals may further cause them to change their attitudes, intentions, or behaviour in order to remain in keeping with perceived group norms, especially in a manner that is similar to decreasing cognitive dissonance [42]. Pluralistic ignorance therefore accords with the influence of minorities [43]. Minority influence, which can be described as the pressure induced by relatively few people, will eventually lead other members to move towards the opinions of the minority [44]. As time passes, minority opinions, which under the influence of pluralistic ignorance are perceived as majority opinions, encourage people to begin to adopt the opinion, especially because this tiny minority of people is behaviourally consistent [44]. Consequently, in consideration of the misleading cognition derived from pluralistic ignorance, an individual may adjust his/her attitudes and behaviours to be more in line with the group's norms. Hence, it is hypothesised that:

**Hypothesis (H3).** *Pluralistic ignorance positively affects employees' subjective norms in an organisation.*

*2.5. Pluralistic Ignorance and Employees' Pro-Environmental Behaviour*

Considering the importance of the impact of misperceived social norms, it is imperative to know the extent of pluralistic ignorance of employees' pro-environmental behaviour. A prior study conducted by Smit-Simone et al. [45] found that pluralistic ignorance played a decisive role in increasing the consumption of a given tobacco product. Thus, an individual's mistakenly perceived social norms, derived from pluralistic ignorance, influenced personal attitude and behaviour, subsequently giving rise to overt behaviour contrary to his/her attitude [46]. This is because individuals, to a large extent, tend to hide their differences from groups as they fear that they may be punished implicitly or explicitly by group members if they disagree [43]. As a result, an individual, although extremely reluctant, may eventually adopt the behaviour of others owing to mistakenly perceived social norms. Thus, it is considered here that pluralistic ignorance has a critical impact on individuals' pro-environmental behaviour. This leads to the Hypothesis H4.

**Hypothesis (H4).** *Pluralistic ignorance positively affects employees' pro-environmental behaviour in an organisation.*

*2.6. The Mediating Role of Pluralistic Ignorance in the Relationship Between Social Identity and Pluralistic Ignorance*

Social identity theory was initially conceptualised by Tajfel [47] who described social identity as a person's awareness about the degree to which he/she belonged to a particular social group, together with the emotional meaning and value of this group to him/her. An individual's self-consciousness, because of interpersonal interactions, enables him/her to categorise social stratification and to categorise any others he/she wants to associate with [48]. Hence, the more closely the identity of an individual is related to other identities, the more likely it is that the individual's behavioural choices will be related to other identities [49]. This is because the behaviours of individuals are organised to change a situation and any perceived self-related meanings in order for them to make an agreement with others in terms of identity criteria. Moreover, Miller and McFarland [50] state that situations of pluralistic ignorance occur relatively easily when they allow group members to observe each other; this could therefore establish a strong social identity for group members, especially when they require a degree of professional spirit in such a context [50]. As a result, pluralistic ignorance can be deemed to be a drive that forms in-group social identities [43] and, by contrast, individuals may act reluctantly and passively with a view to maintaining their misperceived notion of consensus and identity within the group [43]. Hence, we consider that the form of social identity of a group of Chinese employees may involve more pluralistic ignorance on account of their desire to keep the identity of group members in their organisation and individuals may be more inclined to support views that are contrary to those they actually hold. As a result, we further consider employees' misperceptions (pluralistic ignorance) mediates the relationship between social identity and employees' pro-environmental behaviour. Hence, this leads to the following hypothesis.

**Hypothesis (H5).** *Pluralistic ignorance mediates social identity and employees' pro-environmental behaviour.*

*2.7. The Mediating Role of Pluralistic Ignorance in the Relationship between Supervisor–Subordinate Guanxi and Employees' Pro-Environmental Behaviour*

*Guanxi* refers to Chinese interpersonal relationships; it is at the core of Chinese society and stems from five *lun* (dyadic relationships) in Confucianism, which is rooted in the thinking of Chinese society. It contains the five types of interpersonal relationships of the Chinese: i.e., father–son, husband–wife, ruler–subject, elder–younger siblings, and friend–friend [51]. The five types of *guanxi* are based on various criteria concerning the treatment and social interaction leading to different levels of interdependent relationships. Normally, *guanxi* refers to highly private interpersonal networks into which the diffusion of external social interpersonal networks is difficult [52]. It refers to the implicit social relationships existing between people based on common interests and has been regarded as a common form of membership or social exchange based on instrumental purposes [53]. In this setting, colleagues' opinions of an individual are considered to be far more important than the individual's own view of him/herself. In addition, if an employee has a strong and tight *guanxi* with someone in an organisation, the supervisor may give him/her a good evaluation to meet and develop the *guanxi* with that significant person for other purposes in any future job [54]. For another example, in a qualitative interview, one interviewee stated vividly that when an employee enjoyed a better *guanxi* with the supervisors, his or her performance evaluation would be given a higher rating by the supervisors in exchange [54]. Hence, when an individual does not feel that he/she can express himself/herself freely, he/she may sacrifice his/her ambition and eventually take a decisive action in order to gain others' approval and improve their *guanxi* in Chinese society. Consequently, an individual might fear losing colleagues' camaraderie because he/she cares a great deal about what others think of him/her. Here, we further consider employees' misperceptions (pluralistic ignorance) to play a mediating role in enhancing the relationship between social identity and employees' pro-environmental behaviour. Therefore, we hypothesise H6 and Figure 1 depicts the concept of our theoretical model.

**Hypothesis (H6).** *Pluralistic ignorance mediates S-S guanxi and employees' pro-environmental behaviour.*

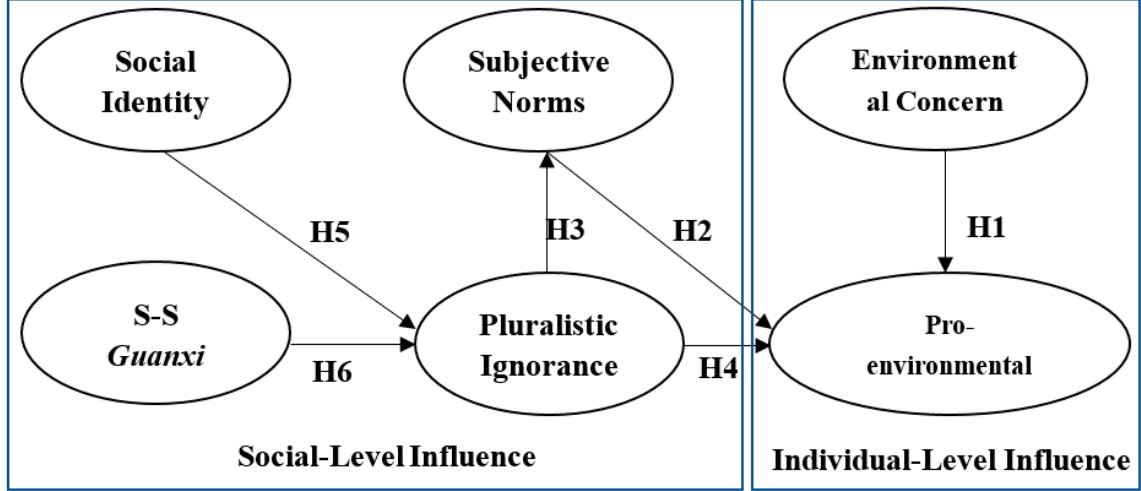

**Figure 1.** Research framework. Note1: H5 tests the mediating effect of pluralistic ignorance on the relationship between social identity and pro-environmental behaviour; H6 tests the mediating effect of pluralistic ignorance on the relationship between social identity and pro-environmental behaviour.

## 3. Methodology

### 3.1. Instrument Development

The theoretical model of this study consists of six constructs composed of 27 statements, which were developed from previous studies. These were examined by two senior academics from Loughborough University (UK) to reinforce the reliability and validity of the instrument. As a result, four items of the questionnaire were modified since the professors thought that the wording and phrasing of these items might be ambiguous. Afterwards, a pilot study was carried out; the questionnaires were gathered initially via email with the effective samples of the pilot numbering 103. The values of individual item reliability (all above 0.7), validity (e.g., Average Variance Extracted (AVEs) ranged from 0.713 to 0.896) and internal consistency (e.g., Cronbach's $\alpha$ ranged from 0.812 to 0.973l, and the composite reliability (CRs) ranged from 0.812 to 0.963) of the measurement model were robust.

### 3.2. Data Connection

The full-scale questionnaire of the study was distributed among the respondents, all of whom were working in Jiangsu Province, China, by two researchers between January and May 2019; a convenient sampling method was utilised. Of the 800 surveys distributed, 548 valid questionnaires (a 78.2% response rate) were utilised for analysis. The collected sample showed a reasonable distribution across gender, age, and education. Table 2 lists the detailed sample profile.

**Table 2.** Sample characteristics (*N* = 548).

| Variables | Demographics | Number | Percent |
|---|---|---|---|
| Occupation | State-owned enterprise | 74 | 13.5% |
| | Private enterprise | 352 | 64.2% |
| | Foreign capital or joint venture | 36 | 6.6% |
| | Collective ownership | 10 | 1.8% |
| | Others | 76 | 13.9% |
| Gender | Male | 188 | 34.3% |
| | Female | 360 | 65.7% |
| Education | High school or below | 2 | 0.4% |
| | College | 20 | 3.6% |
| | University | 488 | 89.1% |
| | Master | 20 | 3.6% |
| | Doctor | 18 | 3.3% |
| Age | 21–25 Years | 450 | 82.1% |
| | 26–30 Years | 34 | 6.2% |
| | 31–35 Years | 18 | 3.3% |
| | 36–40 Years | 14 | 2.6% |
| | 41–45 Years | 18 | 3.3% |
| | 46–50 Years | 8 | 1.5% |
| | 51 Years or above | 6 | 1.1% |
| Working Years | 1 Year or below | 70 | 12.8% |
| | 1–3 Years | 408 | 74.5% |
| | 4–6 Years | 14 | 2.6% |
| | 7–9 Years | 14 | 2.6% |
| | 10–15 Years | 16 | 2.9% |
| | 16–20 Years | 8 | 1.5% |
| | 20 Years or above | 18 | 3.3% |

## 4. Analysis Method

### 4.1. Common on Method Bias

The survey of this study was designed and administered carefully to eliminate the possibility of common method bias. For instance, both positive and negative items were adopted, and the research name of each variable was concealed. Respondents were also required to answer the survey honestly. Meanwhile, Haman's single factor tests were also conducted to examine potential common method bias in this study. When using self-reported surveys to collect data, common method variance may occur if respondents answer perceptual measure concerning explanatory and dependent variables and tend to provide consistent answers to other relevant survey questions [55]. The Haman single factor test showed that the maximum variance of individual factor explanation was 46.17%. In other words, a single factor neither appears nor explains most of the differences between measurements [56].

### 4.2. Measurement Model

Partial least square (PLS) with Smart PLS (2.0.M3), based on structural equational modelling (SEM) [57] was the primary statistically analytic tool utilised in this study while the measurement model included an analysis of reliability and validity. First, each individual item reliability was tested (factor loading), together with the internal consistency of constructs, by evaluating Cronbach's α and composite reliability (CR). Each individual item of the questionnaire showed a value greater than 0.7. The results also showed that Cronbach's α ranged from 0.847 (environmental concern) to 0.982 (pluralistic ignorance). The CRs ranged from 0.897 (environmental concern) to 0.985 (pluralistic ignorance) and the AVE values ranged from 0.686 (environmental concern) to 0.919 (pluralistic ignorance). All the constructs therefore showed good values of internal consistency. Table 3 below shows the relevant values of the measurement model (including skewness and kurtosis).

**Table 3.** The measurement model of the research (all items described are listed in Appendix A).

| Constructs | Indicators | Cronbach's $\alpha$ | M (SD) | Loadings | t-Value | Skewness | Kurtosis |
|---|---|---|---|---|---|---|---|
| Pluralistic Ignorance (CR = 0.985, AVE = 0.919) | PI1 | 0.982 | 3.78 (1.02) | 0.938 *** | 46.9 | −0.40 | −0.29 |
| | PI2 | | 3.81 (0.98) | 0.958 *** | 72.3 | −0.39 | −0.19 |
| | PI3 | | 3.81 (0.98) | 0.966 *** | 67.2 | −0.40 | −0.17 |
| | PI4 | | 3.81 (0.97) | 0.971 *** | 98.2 | −0.36 | −0.28 |
| | PI5 | | 3.80 (0.98) | 0.959 *** | 60.4 | −0.41 | −0.15 |
| | PI6 | | 3.78 (0.99) | 0.960 *** | 63.1 | −0.37 | −0.27 |
| Supervisor–subordinate *Guanxi* (CR = 0.938, AVE = 0.716) | SSG1 | 0.921 | 3.61 (1.08) | 0.897 *** | 35.6 | −0.30 | −0.50 |
| | SSG2 | | 3.61 (1.01) | 0.894 *** | 35.3 | −0.16 | −0.55 |
| | SSG3 | | 3.72 (1.06) | 0.858 *** | 25.1 | −0.30 | −0.63 |
| | SSG4 | | 2.92 (1.32) | 0.779 *** | 14.7 | 0.10 | −1.02 |
| | SSG5 | | 3.35 (1.18) | 0.821 *** | 15.4 | −0.15 | −0.79 |
| | SSG6 | | 3.23 (1.23) | 0.823 *** | 16.5 | −0.19 | −0.84 |
| Social Identity (CR = 0.953, AVE = 0.835) | SI1 | 0.934 | 4.22 (0.99) | 0.939 *** | 53.5 | −1.19 | 0.95 |
| | SI2 | | 4.01 (1.06) | 0.934 *** | 47.4 | −0.83 | 0.16 |
| | SI3 | | 4.27 (0.97) | 0.907 *** | 32.3 | −1.26 | 1.14 |
| | SI4 | | 3.92 (1.14) | 0.876 *** | 29.0 | −0.72 | 0.40 |
| Subjective Norms (CR = 0.956, AVE = 0.846) | SN1 | 0.939 | 3.80 (1.05) | 0.895 *** | 28.7 | −0.53 | −0.39 |
| | SN2 | | 3.80 (1.02) | 0.934 *** | 49.8 | −0.49 | −0.27 |
| | SN3 | | 3.89 (1.04) | 0.941 *** | 51.3 | −0.57 | −0.40 |
| | SN4 | | 3.80 (1.06) | 0.909 *** | 38.3 | −0.64 | −0.01 |
| Environmental Concern (CR = 0.897, AVE = 0.686) | EC1 | 0.847 | 4.03 (1.05) | 0.857 *** | 16.8 | −0.92 | 0.30 |
| | EC2 | | 4.08 (1.07) | 0.842 *** | 16.4 | −1.06 | 0.45 |
| | EC3 | | 3.47 (1.18) | 0.745 *** | 10.5 | −0.17 | −0.96 |
| | EC4 | | 3.39 (1.29) | 0.865 *** | 27.8 | −0.54 | 1.55 |
| Pro-environmental Behaviour (CR = 0.960, AVE = 0.890) | PEB1 | 0.938 | 3.69 (1.02) | 0.931 *** | 88.1 | −0.34 | −0.35 |
| | PEB2 | | 3.69 (0.99) | 0.953 *** | 112 | −0.27 | −0.41 |
| | PEB3 | | 3.79 (0.94) | 0.946 *** | 40.6 | −0.34 | −0.24 |

Note 1. M = mean, SD = standard deviation, CR = composite reliability, SSG = supervisor–subordinate *Guanxi*, PI = pluralistic ignorance, SI = social identity, SN = subjective norms, PEB = pro-environmental behaviour; EC = Environmental Concern. Note 2. Skewness should be less than 2.0. Note 3. Kurtosis should be less than 7.0. *** $p < 0.001$.

The study then tested the discriminant and convergent validities of the research constructs. With regard to convergent validity, the AVEs ranged from 0.686 to 0.919. Next, discriminant validity was evaluated by analysing the square roots of AVE and cross loading. The results showed that all the square roots of AVE were higher than correlations among the constructs. It was also found that the cross loadings of each construct were higher than those of other constructs (as shown in Appendix B). These demonstrated excellent discriminant validity [58]. Table 4 below lists the relevant values of the square root of AVE.

**Table 4.** The diagonal elements show the square root of AVE of each latent construct.

| | Items | Mean/SD | EC | PEB | PI | SSG | SI | SN |
|---|---|---|---|---|---|---|---|---|
| EC | 4 | 3.89 (0.90) | **0.828** | | | | | |
| PEB | 3 | 3.72 (0.93) | 0.570 | **0.943** | | | | |
| PI | 6 | 3.80 (0.95) | 0.615 | 0.740 | **0.958** | | | |
| SSG | 6 | 3.40 (0.97) | 0.538 | 0.599 | 0.679 | **0.846** | | |
| SI | 4 | 4.10 (0.95) | 0.691 | 0.612 | 0.747 | 0.604 | **0.913** | |
| SN | 4 | 3.82 (0.96) | 0.668 | 0.662 | 0.714 | 0.599 | 0.711 | **0.919** |

Note: SD = Standard Deviation.

### 4.3. Structural Model

In Figure 2, the study presents the structural model. First, the path coefficient between environmental concern and employees' pro-environmental behaviour was not significant (β = 0.981, *p* > 0.05). In other words, environmental concern does not necessarily reduce employees' pro-environmental behaviour. Hence, H1 was not supported. We also found a positive and significant relationship between subjective norms and employees' pro-environmental behaviour, which is in support of H2 (β = 0.226, *p* < 0.05). H3 validated the notion that pluralistic ignorance positively increased employees' subjective norms, showing that when employees feel a higher level of pluralistic ignorance, they are more likely to perceive higher subjective norms (β = 0.715, *p* < 0.001). H4 supports the notion that, when there is higher pluralistic ignorance, there will be more subjective norms (β = 0.517, *p* < 0.001).

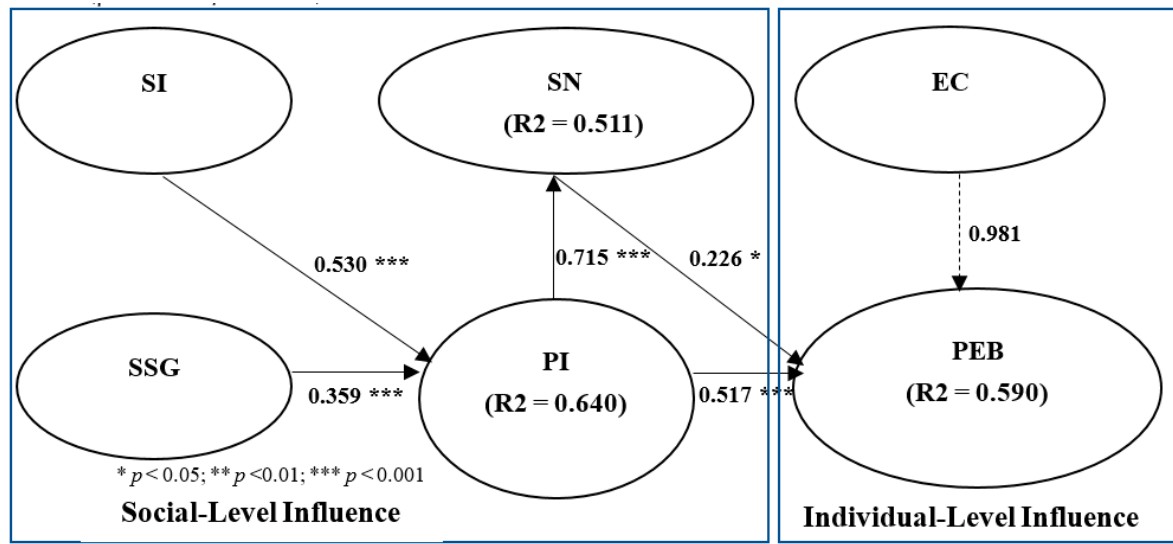

**Figure 2.** Parameter estimates for the structural model. Note: SSG = supervisor–subordinate Guanxi, PI = pluralistic ignorance, SI = social identity, SN = subjective norms, PEB = pro-environmental behaviour; EC=Environmental Concern.

### 4.4. Pluralistic Ignorance as a Mediating Effect

This study employed SmartPLS 2.0.M3 [59] and PROCESS macro 3.4 version [60] for the mediation analysis. As shown in Table 4, the total effect of supervisor–subordinate *Guanxi* (SSG) on pro-environmental behaviour (PEB) is significant (c = 0.598 ***). The results with regard to path α (SSG → pluralistic ignorance (PI): 680 ***) and path β (PI → PEB: 620 ***) are also significant. However, the direct impact of SSG on PEB is shown to be insignificant when PI is included. A key condition to measure a mediating effect is to examine the indirect effect (α × β), which is significant. As mentioned above, we assumed that PI fully mediates the relationship between SSG and PEB (α × β = 0.421), which is in support of H7 [61]. The same approach was utilised to analyse H8. We tested the total effect of social identity (SI) on PEB, which is significant (c = 0.613 ***). Furthermore, we tested path α (SI → PI: 0.747 ***) and path β (PI → PEB: 0.641 ***); both are significant. However, the direct impact of SI on PEB is not significant when containing PI. Hence, PI can be regarded as a full mediation between SI and PEB, in support of H8 [61]. Then, PROCESS Macro with bootstrapping (based on 5000 estimates being taken) was chosen to analyse the mediating effect of this study, resulting in 95% confidence intervals (percentile) for the mediators. If the confidence interval of the indirect effect does not contain 0, the mediating effect is considered to be significant. Moreover, we computed the variance explained for the variance accounted for (VAF) index; this determines the magnitude of the indirect effect (α × β) relative to the total effect (c) [62]. The VAF values are between 20% and 80%, which shows a partial mediation

while the values of VAF which are greater than 80% show full mediation. The results presented in this study show that VAFs for the indirect effect were 81.1% and 80.7%, respectively (Table 5). These results support H7 and H8 (both showing full mediation). Also, Table 6 shows the results of hypothesis testing and P value.

**Table 5.** Results of mediating effect.

| IV | MV | DV | $c$ | $\alpha$ | $\beta$ | $c'$ | $\alpha\beta$ | Percentile Bootstrap 95% Confidence Interval | | VAF | Type |
|---|---|---|---|---|---|---|---|---|---|---|---|
| | | | | | | | | Lower | Upper | | |
| SSG | PI | PEB | 0.598 (0.001) | 0.680 (0.001) | 0.620 (0.001) | 0.098 (n.s.) | 0.421 | 0.050 | 0.290 | 81.1 | Full |
| SI | PI | PEB | 0.613 (0.001) | 0.747 (0.001) | 0.641 (0.001) | 0.114 (n.s.) | 0.478 | 0.052 | 0.210 | 80.7 | Full |

Note 1: IV = independent variable, M = mediating variable, DV = dependent variable. Note 2: SSG = supervisor–subordinate guanxi, SI = social identity, CM = collectivism, PI = pluralistic ignorance. Note 3: c = The total direct effect of the IV on the DV; $\alpha$ = The effect of the IV on the MV; $\beta$ = The effect of the MV on the DV when controlling for the IV; c' = The effect of the IV on the DV when controlling for the MV; $\alpha\beta$ = The total indirect effect on the DV. Note4. 5000 bootstrap samples. (based on t (4999), one-tailed test). t (0.05, 4999) = 1.64; t (0.01, 4999) = 2.32; t (0.001, 4999) = 3.09.

**Table 6.** Results of hypothesis testing and *P* value.

| Hypotheses | Result |
|---|---|
| **H1.** Environmental concern positively affects employees' pro-environmental behaviour in an organisation | Not Supported |
| **H2.** Social norms of employees positively affect their pro-environmental behaviour in an organisation. | Supported * |
| **H3.** Pluralistic ignorance positively affects employees' subjective norms in an organisation. | Supported *** |
| **H4.** Pluralistic ignorance positively affects employees' pro-environmental behaviour in an organisation. | Supported *** |
| **H5.** Pluralistic ignorance mediates social identity and employees' pro-environmental behaviour. | Supported (Full Mediation) |
| **H6.** Pluralistic ignorance mediates S-S *guanxi* and employees' pro-environmental behaviour. | Supported (Full Mediation) |

Note: * $p < 0.05$; *** $p < 0.001$.

*4.5. Differential Analysis of Demographic Variables*

Having considered that gender, age, working experience, and occupation might affect employees' pro-environmental behaviour, the research carried out an independent-sample t test and one-way ANOVA test. The results showed that there were no significant impacts on the study's main constructs with regard to the abovementioned variables (all analytic results of the ANOVA are listed in Appendix C, Appendix D, Appendix E, and Appendix F).

## 5. Discussion and Conclusions

This study makes numerous practical and theoretical contributions to the domain of environmental management, especially with regard to employees' pro-environmental behaviour. These are as follows.

*5.1. Theoretical Implications*

First, in view of the lack of previous relevant research, this study makes use of the concept of misperceived and multiple social norms to build a theoretical model to offer an advanced understanding of the causes and effects of pluralistic ignorance on the intertwined relationships among supervisor–subordinate *guanxi*, social identity, subjective norms, and employees' pro-environmental behaviour within the complex context of contemporary Chinese organisations. Likewise, prior studies seem not to have examined how supervisor–subordinate *guanxi*, social identity, and subjective norms are interwoven and associated with pluralistic ignorance which, in turn, impacts employees' pro-environmental behaviour. Therefore, this study offers a novel model for environmental management regarding pro-environmental behaviour, especially in the collectively cultural context of Chinese society.

Furthermore, the results show that the social processes (i.e., supervisor–subordinate *guanxi* and social identity) respectively play critical roles in affecting employees' pluralistic ignorance which, in turn, influences their pro-environmental behaviour within organisations in Chinese society. This is, in all probability, because Chinese employees believe that having a fundamental connection with important people (e.g., supervisors) is extremely important in organisations and consider that the cultivation of social harmony and cooperation is relatively important for them [51,63]. In line with this, we thus consider that social determinants are exerting a subtle influence on Chinese employees' judgements and behaviours in such a cultural context. Our analytic results show that increasing pluralistic ignorance cannot only bridge the gap between supervisor–subordinate *guanxi* and employees' pro-environmental behaviour but also connect the relationship between social identity and their pro-environmental behaviour. It seems that when individuals feel that they are unable to express themselves freely, they may sacrifice their own ambition and finally take decisive action to identify with others and improve their *guanxi* in Chinese society. In short, Chinese employees' pluralistic ignorance can improve their pro-environmental behaviour in this collective cultural setting.

In addition, our findings show that employees' environmental concerns do not have an impact on their pro-environmental behaviour in organisations whereas many relevant individual psychological theories reveal that concerns can effectively explain the process in terms of human decision-making processes in terms of pro-environmental behaviour [35,64,65]. This insignificant relationship may be attributed to a greater or lesser extent to voluntary and persuasive characteristics of such pro-environmental behaviour. Hence, if managers or supervisors can impose more or fewer restrictions on performing non-environmental behaviour in organisations, employees may increase their willingness to carry out such pro-environmental behaviour in organisations in a contemporary Chinese society. We have thus examined the role of misperceived social norms (i.e., pluralistic ignorance) in Chinese campuses and have added to the body of knowledge that suggests considering both correct conceptions and misperceptions of social norms can account for the insufficiency of the theory in the context of contemporary Chinese society.

Third, we extensively explore the gap between social influence (e.g., subjective norms) and employees' pluralistic ignorance in a Chinese organisation. Although social influence has been utilised pervasively to explain a vast range of pro-environmental behaviour, few studies seem to have investigated the influences of misperceived social norms on employees' subjective norms within the highly collective context of Chinese organisations. Our results confirm that employees' pluralistic ignorance positively affects their subjective norms; this means that Chinese people tend to be more concerned about the feelings of other employees and this leads them to their final decision-making, especially in the context of the collective culture.

Finally, this study attempts to understand the influence of environmental concern of employees' intrinsic motivation on their pro-environmental behaviour in the context of Chinese contemporary culture. The comprehensive model, rather than considering environmental concerns, shows that social psychology and behaviour (i.e., supervisor–subordinate *guanxi*, social identity, and subjective norms) seem to play a more significant role in influencing an employee's pluralistic ignorance within organisations in the context of the Chinese culture. Therefore, when managers of organisations focus

efforts on the pursuit of environmental management, their employees seem to care more about their social network structure and the potential interpersonal relationships within the organisation, rather than their personal environmental concerns.

*5.2. Managerial Implications*

With the ever-accelerating development of economy and rapid deterioration of ecological environment, the Chinese government has made a massive effort in maintaining the environmental equilibrium. However, such efforts worked by the government for environmental protection may be in vain if Chinese citizen pay less attention to concerning their natural surroundings. This study proposes an integrated theoretical framework for improving pro-environmental behaviour by highlighting the critical impact of pluralistic ignorance on managerial and social psychology. The findings extend the current literature advocating individual psychology, social psychology, and pluralistic ignorance of employees, which are the causes and results of employees' ecological behaviour in the contemporary social environment of China. The findings not only make a contribution to the field of environmental management and practice, they also offer new insight for relevant industrial managers who may consider employees' pro-environmental behaviours to achieve their pro-environmental goals as one criterion when selecting appropriate employees.

Also, this novel model offers advice on the importance of employees' pluralistic ignorance under certain circumstances. In light of previous research, which has focused largely on the completely negative impact of pluralistic ignorance on human decision-making processes [24,28,46,66], this study notes that pluralistic ignorance does not necessarily lead to passive consequences. Even though most research on pluralistic ignorance focuses on the adoption of problematic social consensus, we find that pluralistic ignorance, in some cases, may give rise to greater adoption of productive norms and behaviour. For instance, a study concerning the influence of pluralistic ignorance on employees' ethics carried out by Halbesleben et al. [28] showed that employees normally survey others' unethical behaviour initially while often overestimating the level of other employees' unethical behaviour. This results in a sense of "I am more ethical than other employees in the organization" or vice versa. Hence, following this line of thought, our interpretation of pluralistic ignorance is relatively valuable because it enhances our understanding of group decision-making mechanisms in the cultural context of the studied setting. Moreover, to trigger social norms within the organisation, managers should carefully design and implement their ongoing communication strategies because group norms are mainly generated by information and communication among group members in Chinese society; this emphasises that communication with others is part of the collective. This task may be easier to accomplish in an organisation with small segmentation and with target members who are relatively homogeneous.

In addition, this research suggests that managers of organisations should pay attention to giving employees' a sense of identity within the organisation, especially in terms of sub-groups in which they often participate; they should also influence group social norms to cultivate further members' pro-environmental behaviour. In order to develop and further enhance the identity of the organisation's employees, managers should pay more attention to building a favourable image of their organisation's internal and external characteristics. For instance, the responsiveness of supervisors of the organisation to pro-environmental issues makes organisational identity more attractive; it also evokes subordinates' sense of obligation in return for "comradeship", which further promotes the development of psychological ties within the organization, especially in Chinese society. In addition, members may be proud to demonstrate pro-environmental behaviour because they believe that such behaviour has socially valuable characteristics (for example, a positive external image), and their self-esteem can be increased through group achievements and reputations.

More importantly, it was found that pluralistic ignorance fully mediates the relationships between supervisor–subordinate *guanxi*, social identity and employees' pro-environmental behaviour within organisations in a Chinese cultural context. In this setting, the common practices that affect employees'

pro-environmental behaviour include enhancing the external benefits gained by the contributing members. For example, public recognition of contributions provides an important social reward; this gives clear value to members' contributions and increases the perceived meaningfulness of active participation. However, this type of incentive mechanism needs to be treated with caution because excessive use of external incentive systems in organisations may backfire. Public recognition may inadvertently convey a sense of superiority to those who contribute the effort, not to those who receive it. As a result, public recognition can have a devastating effect in collective efforts where people prefer integration rather than prominence.

Rather than merely considering an individual's intrinsic motivation and behaviour traditionally, this study conceptualises the contributions to organisations that arise because of the will of the individual, with group behaviours and actions being the result. This study clarifies the essence of the psychology of an individual, which focuses exclusively on personal intentions because the research shows that when a person intends to be part of a group activity, he or she may have a collective intention. This kind of conceptualisation is particularly important in organisations based on the Chinese cultural context in which social interaction is the most important factor in targeting and attracting individual participants. Therefore, the pluralistic ignorance of an individual, encapsulated in collective members' common behaviour, is puzzling and should be measured by researchers who want to make accurate predictions or inferences about a group's intentions and/or behaviour. This study also extends previous ones to demonstrate that pluralistic ignorance fully mediates the relationships between social identity, supervisor–subordinate *guanxi*, and employees' pro-environmental behaviour.

## 6. Conclusions

The analytic results in this study concerning pluralistic ignorance bring to light numerous critical issues, which remain to be discussed in more depth. Misperceived social norms describe the gap between an individual's real attitude or behaviour and how he/she actually thinks about the attitudes or behaviours of others. In the meantime, pluralistic ignorance is a social phenomenon regarding the most common misperception that occurs when a person has an individual view of him/herself and mistakenly assumes that the overwhelming majority of the group holds the opposite view [19]. Pluralistic ignorance is widely applied to a variety of psychological and social environments concerning risk behaviour, including social projection [20], community values [23], virtual community [26], and drug users [27]. In light of the many research studies on pluralistic ignorance, which have focused on the adaptation of problematic social norms, this study found that pluralistic ignorance in some cases can give rise to the adoption of positive and productive norms in terms of employees' pro-environmental behaviour in the collective spirit of Chinese society.

With the analytic results of this study, we find pluralistic ignorance is a strong proximal determinant of social norms, playing a key mediating role in connecting the relationships among social identity, supervisor–subordinate guanxi, and pro-environmental behaviour. We also find that subjective norms contribute to pro-environmental behaviour, while environmental concern seems to receive less consideration from employees in a Chinese society. Hence, pluralistic ignorance can also be positive if employees' predisposition to improve supervisor–subordinate *guanxi* ultimately modifies the course of action. Meanwhile, in a Chinese cultural context, it can be imagined that it is actually the best practice for an employee to support his/her supervisors' opinions (which, through multiple ignorance, can become a majority opinion), perhaps because such people are considered to have greater insight and experience. Common organisational development interventions include evaluation and feedback; these can assist organisations in recognising and combatting widescale ignorance and can also assist in shedding light on the potential disconnections between the actual opinions of employees and norms across the organisation. Of course, this assumes that employees' views have not changed due to multiple ignorance. This intervention can be particularly useful when multiple ignorance leads to consistency with powerful members of the organisation, such as managers, because employees may feel that they have publicly to obey that person. An employee may also feel that he/she can be

outspoken and offer his/her true thoughts on an evaluation so as to have a clearer understanding of the organisation's norms. As a result, these people may listen to their supervisors and obtain better results.

In addition, the evidence analysed by this research shows that pluralistic ignorance offers lucid social cognitive mechanisms that bridge individual-level and social-level psychologies to understand comprehensively how social information processes affect the outcome of group decision-making. Specifically, it provides clearer process guidance than groupthink. For example, the model of multiple social norms and pluralistic ignorance employed here explains why negative fantasies, created by perceived threats to supervisor–subordinate *guanxi* and social identity, can be created. Furthermore, one characteristic of groupthink is that various aspects of information processing can be clearly explained, especially the consideration of various options, through an understanding of the impact of a few. What is most remarkable is that pluralistic ignorance exceeds groupthink on account of its critical impact on multiple phenomena at individual, group, and organisational levels, especially in a Chinese cultural context where, if an individual assumes that others will conform and that others also want him/her to conform, he/she may be more likely to undertake pro-environmental behaviour.

Consequently, this study highlights an important circumstance that prior researchers seem to have largely ignored although it has many important implications for organisations. Thus, this article bridges the relationship between management and key concepts in the integration of individual psychology, social psychology, and environmental psychology. The multi-level concept applied by the study, interestingly, assists in promoting the integration of psychological and management concepts within a group environment. Further research may embed and consider multiple ignorance from a theoretical view in an organisational background as such theory would enable researchers to transcend the application of new examples of multiple ignorance in organisations, as well as provide a framework for the study of these new applications.

Despite our study's novel findings, this work still has several limitations that need to be further investigated in the future. First, we cannot confirm whether our findings could be applied to all regions and types of organisations in contemporary Chinese society because of the limited sample of Jiangsu Province, China. Further research could increase the population under scrutiny and follow outcomes with different stages of samples. Second, this study sets up pluralistic ignorance as misperceived social norms at an individual level. Future research may further explore pluralistic ignorance at a group-level perspective. Finally, even though this study has validated that social identity and subjective norms successfully explain employees' pluralistic ignorance through the cross-sectional combination of a quantitative approach, this may not capture dynamic behaviour of the formation of employees' pluralistic ignorance. Hence, this study suggests that future research might consider utilising a qualitative approach and longitudinal data collection in order to explore deeply the impact of pluralistic ignorance on pro-environmental behaviour from a dynamic perspective.

**Author Contributions:** H.-F.C. developed the theoretical model and analysed the quantitative data. J.-W.S. and K.-J.S. performed the research and collected the data during the period. All authors have read and agreed to the published version of the manuscript.

**Funding:** This research was supported by a project (2018R091) from the Startup Foundation for Introducing Talent of NUIST (Nanjing University of Information Science and Technology).

**Conflicts of Interest:** The authors declare no conflict of interest.

## Appendix A

**Table A1.** Measurement instrument (Notes. Items marked (-) are reverse scored).

| Factor | Items | Author(s) |
|---|---|---|
| **Social identity**: The degree to which employees identify with other colleagues in the organisation. | | |
| SI1 | I regard myself as an important member of the organisation. | Kowert and Oldmeadow [67] |
| SI2 | I am not pleased to be a member of the organisation (-). | |
| SI3 | I identify with my colleagues in the organisation. | |
| SI4 | I feel strong ties with my colleagues of the organisation. | |
| **Subjective norms**: The degree to which employees involve in subjective norms in the organisation. | | |
| SN1 | My colleagues think acting in a pro-environmental manner is important. | Zhang et al. [68] |
| SN2 | My colleagues who influence my decisions think I should act favourably towards the natural environment. | |
| SN3 | Most colleagues who are important to me consider that engaging in pro-environmental behaviour is desirable. | |
| **Supervisor–subordinate *guanxi***: The degree to which an employee is involved in Supervisor–subordinate *guanxi* in the organisation. | | |
| SSG1 | I will call my supervisors or visit them after office hours. | Law et al. [69]; Guan and Frenkel [63] |
| SSG 2 | Normally, my supervisor invites me to his/her home for the meal. | |
| SSG 3 | When my supervisor's birthday comes, I will visit and send him/her gifts. | |
| SSG 4 | I don't share my thoughts and feelings with my supervisor (-). | |
| SSG 5 | I care about my supervisor's family and work conditions. | |
| SSG6 | Whenever I will stand on my supervisor's side. | |
| **Pluralistic ignorance**: The level of employees' pluralistic ignorance, especially the resulting perception of difference in an organisation. | | |
| PI1 | Colleagues are willing to perform pro-environmental behaviour in the organisation. | Soroa-Koury and Yang [18] |
| PI2 | My colleagues and I both perform pro-environmental behaviour in the organisation. | |
| PI3 | Colleagues are willing to perform pro-environmental behaviour in activities of the organisation. | |
| PI4 | My colleagues and I both perform pro-environmental behaviour in activities of the organisation. | |
| PI5 | Colleagues will continue to perform pro-environmental behaviour in the organisation in the future. | |
| PI6 | My colleagues and I will continue to perform pro-environmental behaviour in the organisation in the future. | |
| **Environmental concern**: The extent to which employees express their concerns for the environment. | | |
| EC1 | I don't think we've done enough to protect our natural resources. | Yusof et al. [64]; Chen and Tung [70] |
| EC2 | I'm sorry that the government didn't take more measures to control the environmental pollution. | |
| EC3 | People pay more attention to air and water pollution than is reasonable. | |
| EC4 | When I think about the harm pollution does to plants and animals, I feel angry and depressed | |
| **Pro-environmental behaviour**: The extent to which employees perform pro-environmental behaviour. | | |
| PEB1 | When I leave the office, I turn off the light. | De Leeuw et al. [71] |
| PEB2 | At the office, I tend not to put my trash in the proper recycling bin (-). | |
| PEB3 | I turn off electronic appliances when I go eat outside office. | |

# Appendix B

**Table A2.** Cross loadings of each construct.

|        | EC        | SN        | SSG       | SI        | PEB       | PI        |
|--------|-----------|-----------|-----------|-----------|-----------|-----------|
| EC1    | **0.856** | 0.564     | 0.456     | 0.616     | 0.432     | 0.512     |
| EC2    | **0.842** | 0.560     | 0.432     | 0.557     | 0.467     | 0.526     |
| EC3    | **0.746** | 0.444     | 0.401     | 0.402     | 0.396     | 0.369     |
| EC4    | **0.864** | 0.625     | 0.478     | 0.679     | 0.567     | 0.601     |
| SN1    | 0.623     | **0.894** | 0.553     | 0.634     | 0.598     | 0.630     |
| SN2    | 0.628     | **0.933** | 0.560     | 0.671     | 0.613     | 0.666     |
| SN3    | 0.601     | **0.940** | 0.530     | 0.668     | 0.608     | 0.654     |
| SN4    | 0.607     | **0.909** | 0.561     | 0.643     | 0.617     | 0.676     |
| SSG1   | 0.506     | 0.575     | **0.896** | 0.572     | 0.561     | 0.649     |
| SSG2   | 0.534     | 0.590     | **0.894** | 0.614     | 0.586     | 0.677     |
| SSG3   | 0.526     | 0.535     | **0.858** | 0.618     | 0.575     | 0.657     |
| SSG4   | 0.334     | 0.365     | **0.779** | 0.325     | 0.376     | 0.401     |
| SSG5   | 0.391     | 0.484     | **0.820** | 0.439     | 0.439     | 0.510     |
| SSG6   | 0.367     | 0.429     | **0.823** | 0.395     | 0.434     | 0.460     |
| SI1    | 0.673     | 0.678     | 0.558     | **0.938** | 0.559     | 0.704     |
| SI2    | 0.603     | 0.639     | 0.542     | **0.933** | 0.566     | 0.681     |
| SI3    | 0.670     | 0.686     | 0.506     | **0.907** | 0.544     | 0.672     |
| SI4    | 0.579     | 0.595     | 0.603     | **0.876** | 0.569     | 0.673     |
| PEB1   | 0.553     | 0.618     | 0.548     | 0.566     | **0.930** | 0.655     |
| PEB2   | 0.530     | 0.625     | 0.585     | 0.577     | **0.953** | 0.712     |
| PEB3   | 0.532     | 0.631     | 0.561     | 0.588     | **0.945** | 0.725     |
| PI1    | 0.319     | 0.689     | 0.656     | 0.707     | 0.710     | **0.938** |
| PI2    | 0.236     | 0.688     | 0.653     | 0.707     | 0.713     | **0.957** |
| PI3    | 0.434     | 0.679     | 0.657     | 0.714     | 0.719     | **0.966** |
| PI4    | 0.371     | 0.672     | 0.650     | 0.727     | 0.705     | **0.971** |
| PI5    | 0.229     | 0.707     | 0.625     | 0.735     | 0.719     | **0.959** |
| PI6    | 0.376     | 0.674     | 0.666     | 0.707     | 0.691     | **0.960** |

# Appendix C

**Table A3.** ANOVA (working experience).

|       |                | Sum of Squares | df  | Mean Square | F     | Sig.  |
|-------|----------------|----------------|-----|-------------|-------|-------|
| SI    | Between Groups | 9.430          | 6   | 1.572       | 1.741 | 0.109 |
|       | Within Groups  | 488.450        | 541 | 0.903       |       |       |
|       | Total          | 497.880        | 547 |             |       |       |
| SN    | Between Groups | 4.396          | 6   | 0.733       | .792  | 0.576 |
|       | Within Groups  | 500.274        | 541 | 0.925       |       |       |
|       | Total          | 504.670        | 547 |             |       |       |
| BC    | Between Groups | 6.786          | 6   | 1.128       | 1.40  | 0.244 |
|       | Within Groups  | 436.048        | 541 | 0.806       |       |       |
|       | Total          | 442.832        | 547 |             |       |       |
| SSG   | Between Groups | 5.420          | 6   | 1.237       | 1.306 | 0.253 |
|       | Within Groups  | 513.334        | 541 | 0.947       |       |       |
|       | Total          | 518.754        | 547 |             |       |       |
| PI    | Between Groups | 5.891          | 6   | 0.982       | 1.091 | 0.367 |
|       | Within Groups  | 486.995        | 541 | 0.900       |       |       |
|       | Total          | 492.887        | 547 |             |       |       |
| PEB   | Between Groups | 5.091          | 6   | 0.848       | 0.972 | 0.443 |
|       | Within Groups  | 472.156        | 541 | 0.873       |       |       |
|       | Total          | 477.247        | 547 |             |       |       |

## Appendix D

**Table A4.** ANOVA (age).

|  |  | Sum of Squares | df | Mean Square | F | Sig. |
|---|---|---|---|---|---|---|
| SI | Between Groups | 7.411 | 6 | 1.235 | 1.362 | 0.228 |
|  | Within Groups | 490.469 | 541 | 0.907 |  |  |
|  | Total | 497.880 | 547 |  |  |  |
| SN | Between Groups | 7.32 | 6 | 1.220 | 1.337 | 0.236 |
|  | Within Groups | 493.392 | 541 | 0.912 |  |  |
|  | Total | 500.712 | 547 |  |  |  |
| BC | Between Groups | 8.64 | 6 | 1.440 | 1.832 | 0.120 |
|  | Within Groups | 425.226 | 541 | 0.786 |  |  |
|  | Total | 433.866 | 547 |  |  |  |
| SSG | Between Groups | 6.152 | 6 | 1.025 | 1.080 | 0.373 |
|  | Within Groups | 513.601 | 541 | 0.949 |  |  |
|  | Total | 519.754 | 547 |  |  |  |
| PI | Between Groups | 5.996 | 6 | 0.999 | 1.110 | 0.355 |
|  | Within Groups | 486.890 | 541 | 0.900 |  |  |
|  | Total | 492.887 | 547 |  |  |  |
| PEB | Between Groups | 5.298 | 6 | 0.883 | 1.032 | 0.385 |
|  | Within Groups | 462.555 | 541 | 0.855 |  |  |
|  | Total | 467.853 | 547 |  |  |  |

## Appendix E

**Table A5.** ANOVA (educational level).

|  |  | Sum of Squares | df | Mean Square | F | Sig. |
|---|---|---|---|---|---|---|
| SI | Between Groups | 4.086 | 4 | 1.017 | 1.125 | 0.232 |
|  | Within Groups | 490.872 | 543 | 0.904 |  |  |
|  | Total | 494.958 | 547 |  |  |  |
| SN | Between Groups | 2.978 | 4 | 0.744 | .806 | 0.522 |
|  | Within Groups | 501.693 | 543 | 0.924 |  |  |
|  | Total | 504.670 | 547 |  |  |  |
| BC | Between Groups | 5.317 | 4 | 1.329 | 1.639 | 0.208 |
|  | Within Groups | 440.373 | 543 | 0.811 |  |  |
|  | Total | 445.690 | 547 |  |  |  |
| SSG | Between Groups | 3.088 | 4 | 1.772 | 1.936 | 0.158 |
|  | Within Groups | 506.666 | 543 | 0.915 |  |  |
|  | Total | 509.754 | 547 |  |  |  |
| PI | Between Groups | 5.732 | 4 | 1.433 | 1.624 | 0.214 |
|  | Within Groups | 478.926 | 543 | 0.882 |  |  |
|  | Total | 484.658 | 547 |  |  |  |
| PEB | Between Groups | 4.755 | 4 | 1.189 | 1.366 | 0.244 |
|  | Within Groups | 472.491 | 543 | 0.870 |  |  |
|  | Total | 477.247 | 547 |  |  |  |

## Appendix F

**Table A6.** ANOVA (unit).

|  |  | Sum of Squares | df | Mean Square | F | Sig. |
|---|---|---|---|---|---|---|
| SI | Between Groups | 3.975 | 4 | 0.994 | 1.092 | 0.359 |
|  | Within Groups | 493.905 | 543 | 0.910 |  |  |
|  | Total | 497.880 | 547 |  |  |  |
| SN | Between Groups | 4.586 | 4 | 1.147 | 1.245 | 0.291 |
|  | Within Groups | 500.084 | 543 | 0.921 |  |  |
|  | Total | 504.670 | 547 |  |  |  |
| BC | Between Groups | 5.072 | 4 | 1.268 | 1.542 | 0.189 |
|  | Within Groups | 446.558 | 543 | 0.822 |  |  |
|  | Total | 451.630 | 547 |  |  |  |
| SSG | Between Groups | 5.512 | 4 | 1.378 | 1.473 | 0.191 |
|  | Within Groups | 507.705 | 543 | 0.935 |  |  |
|  | Total | 513.217 | 547 |  |  |  |
| PI | Between Groups | 5.076 | 4 | 1.269 | 1.419 | 0.231 |
|  | Within Groups | 485.442 | 543 | 0.894 |  |  |
|  | Total | 490.518 | 547 |  |  |  |
| PEB | Between Groups | .858 | 4 | 0.215 | 0.245 | 0.913 |
|  | Within Groups | 476.388 | 543 | 0.877 |  |  |
|  | Total | 477.247 | 547 |  |  |  |

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
