# Peer review of "Why Employees Contribute to Pro-Environmental Behaviour: The Role of Pluralistic Ignorance in Chinese Society"

_sustainability, doi:10.3390/su12010239_

Round 1
Reviewer 1 Report
The manuscript entitled "Why employees contribute to pro-environmental behavior: the role of pluralistic ignorance in Chinese society" presents a topic of interest and, in general, its proposals are well founded. However, it presents an essential methodological problem that seems necessary to address.
At the theoretical level, the authors do not substantiate the two mediation effects sufficiently. Besides, the paper poses two moderation effects. The first mediation effect refers to H7. "H7. Pluralistic ignorance, especially the resulting perception of difference, mediates social identity and employees 'pro-environmental behavior. ” For there to be a mediation effect of Pluralistic ignorance between social identity and employees' pro-environmental behavior there must be a direct effect between Pluralistic ignorance and pro-environmental behavior. This effect is not observed in the manuscript. Besides, if a medication effect were proposed, it would not be necessary to state hypotheses H2 and H4.
The second mediation effect refers to H8. "H8. Pluralistic ignorance, especially the resulting perception of difference, mediates SS guanxi and employees 'pro-environmental behavior. ” In the same way, for there to be a mediating effect of Pluralistic ignorance between SS guanxi and employees' pro-environmental behavior, it is necessary that there is a direct effect between SS guanxi and pro-environmental behavior, which is not observed in the paper. In addition, if a medication effect were raised, it would not be necessary to raise hypotheses H1 and H3.
On the other hand, to statistically address the effect of moderation, you can consult the article Felipe, C.M., Roldán, J.L., & Leal-Rodríguez, A.L. (2016). An explanatory and predictive model for organizational agility. Journal of Business Research, 69, 4624–4631.
Other aspects that could be addressed.
Introduction
The objectives could be on page 2. All the objectives could be considered as “analyse.”
The hypotheses (H1, H2, H3, and H4) are complicated to read.
Methodology
3.1. Instrument development
Present the data from the pilot study.
Discussion and conclusion
5.1. Theoretical implications
The authors should make an effort to present references at this point. Primarily that was studied with literature and the contribution of this work.
5.2. Managerial implications
Likewise, authors should make an effort to present bibliographical references at this point. Primarily that was studied with literature and the contribution of this work to the management of organizations.
Conclusion
Perhaps this point would be 5.3. Conclusion
The limitations of work and future lines of research should be presented.
Reviewer 2 Report
This paper aims at exploring the employee’s pro-environmental behaviour in the workplace, in the Chinese context. Partial Least Squares (PLS), three-step method and ANOVA tests are proposed as quantitative analyses.
The research question is rather innovative, but it is not well defined and presented in quite a confusing way; this paper, therefore, presents some inaccuracies that have to be addressed and corrected before publication.
Originality/Novelty: Is the question original and well defined? Do the results provide an advance in current knowledge?
The question is original, but it is not completely defined. Results provide interesting remarks, but their discussion seems incomplete.
Significance: Are the results interpreted appropriately? Are they significant? Are all conclusions justified and supported by the results? Are hypotheses and speculations carefully identified as such?
Results are not always proper commented, and conclusions are rather repetitive. Some speculations are nor carefully identified (see comments below).
Quality of Presentation: Is the article written in an appropriate way? Are the data and analyses presented appropriately? Are the highest standards for the presentation of the results used?
The article is written in standard English; some parts concerning the methods and the results are not well commented.
Scientific Soundness: is the study correctly designed and technically sound? Are the analyses performed with the highest technical standards? Are the data robust enough to draw the conclusions? Are the methods, tools, software, and reagents described with sufficient details to allow another researcher to reproduce the results?
The study seems correctly designed, but the general construct seems too complicated. Above all, data refers to a case study that does not allow drawing a general conclusion for the entire Chinese context.
Interest to the Readers: Are the conclusions interesting for the readership of the Journal? Will the paper attract a wide readership, or be of interest only to a limited number of people? (please see the Aims and Scope of the journal)
Conclusions could be interesting for the readership of the Journal, after a proper revision.
Overall Merit: Is there an overall benefit to publishing this work? Does the work provide an advance towards the current knowledge? Do the authors have addressed an important long-standing question with smart experiments?
Not completely.
English Level: Is the English language appropriate and understandable?
The article is written in standard English
Review Comments to the Author
Dear Authors, congratulations on submitting a welcome contribution to this field of study. I have many major requests for expansion and clarification:
MAJOR COMMENTS
Introduction: I appreciate your discussion concerning environmental awareness and corporate pro-environmental behaviour. However, these two parts seem not linked, and I think this section should be rewritten in a clearer way. The paper is based on eight hypotheses; they are maybe too much, and the resulting model is too complicated. In that way, interpreting results is a challenging task.
Methodology: Could you please provide some references for the “27 statements which were developed from previous studies”? Could you please provide a table containing the values of individual item responsibility?
Structural model: Could you provide a table containing the result of the model? Could you also provide a table containing results for the ANOVA test?
Discussion and conclusion: This part contains many repetitions and should be rewritten in a clearer way, highlighting how the hypotheses are supported by the results. Moreover, I found some confusion concerning “pluralistic ignorance”: at first Authors say that it can improve the pro-environmental behaviour, then, under “Managerial implications”, they write that managers should reduce this kind of ignorance under certain circumstances. This argument should be clarified.
Although I think the topic raised in this paper is interesting, the paper should be improved substantively before considering a publication; I cannot recommend publication at this stage. I really appreciate the general presentation of the research, but this article seems a brief recap of a broader thesis, some parts are not clearly linked and some comments seems not properly deepened.
Round 2
Reviewer 1 Report
The authors have made a significant effort to address the suggestions that had been raised. In general, all recommendations have been made; however, the Sobel test is not the most appropriate to demonstrate the effect of mediation, as already indicated in the previous review.
Author Response
Statement Summarising the Revision Process
The authors would like to express their gratitude to the reviewer who has examined this article carefully and provided valuable suggestions. In response to the suggestions made by the reviewer, the researchers have amended and supplemented the content of this article. The detail is as follows:
Q1. The authors have made a significant effort to address the suggestions that had been raised. In general, all recommendations have been made; however, the Sobel test is not the most appropriate to demonstrate the effect of mediation, as already indicated in the previous review.
Answer: In order to enhance the added value and importance of this study, the researchers spent more time reinterpreting and supplementing the outcomes from the mediation in the data analysis of the study. For example, this study adopted PROCESS version 3.4, which utilises percentile bootstrap confidence intervals as the default (bias correction is not available in version 3), to further evaluate the significance of the mediation. The researchers also added the following: “…This study employed SmartPLS 2.0.M3 [59] and PROCESS macro 3.4 version [60] for the mediation analysis. As shown in Table 4, the total effect of SSG on PEB is significant (c=0.598***)…Then, PROCESS Macro with bootstrapping (based on 5000 estimates being taken) was chosen to analyse the mediating effect of this study, resulting in 95% confidence intervals (percentile) for the mediators. If the confidence interval of the indirect effect does not contain 0, the mediating effect is considered to be significant. Moreover, we computed the variance explained for the VAF index; this determines the magnitude of the indirect effect (α × β) relative to the total effect (c) [62] …More details can be seen in Section 4.4: Pluralistic ignorance as a mediating effect, pages 9-10 and Table 4.

Reviewer 2 Report
This paper aims at exploring the employee’s pro-environmental behaviour in the workplace, in the Chinese context. Partial Least Squares (PLS), three-step method and ANOVA tests are proposed as quantitative analyses.
The research question is rather innovative, now it is well defined and presented in a clear way.
Dear Authors, I thank you for the answers to my earlier comments and the changes made in the manuscript (all my questions and requests have been answered). I enjoyed reading your revision (in particular, the revised introduction section and the discussion about “pluralistic ignorance”) and now I feel that this paper will contribute to the field.
Below are some more comments that should guide the paper towards publication:
- Please insert in Figure 1 “H1” (it seems that it is missing);
- Explanations about results are adequate, conclusions need to be addressed just a bit more concerning outcomes from modelling.
Originality/Novelty: I appreciate the authors’ detailed comments on my first review of their work. It has greatly improved with this submission.
Significance: The paper addresses an interesting theme. As such, it can provide some original contribution to the field.
Quality of Presentation: Explanations about results are now adequate.
Scientific Soundness: Authors have polished the “Introduction” section and the “Analysis method” section, with good results in terms of clarity and consistency of argument. Thanks again for having took into account much of the feedback they received.
Interest to the Readers: I think that now the paper is much more interesting for the readers.
Overall Merit: Average
English Level: The article is written in standard English
Author Response
Statement Summarising the Revision Process
The authors would like to express their gratitude to the reviewer who has examined this article carefully and provided valuable suggestions. In response to the suggestions made by the reviewer, the researchers have amended and supplemented the content of this article. The detail is as follows:
Q1. Please insert in Figure 1” H1” (it seems that it is missing).
Answer: As suggested by the reviewer, the researchers have added “H1” to Figure 1 (see Figure 1, page 6).
Q2. Explanations about results are adequate, conclusions need to be addressed just a bit more concerning outcomes from modelling.
Answer: In order to enhance the added value and importance of this study, the researchers spent more time carefully supplementing outcomes of the theoretical model in the conclusion of the study. For example, the researchers added the following: “…With the analytic results of this study, we find pluralistic ignorance is a strong proximal determinant of social norms, playing a key mediating role in connecting the relationships among social identity, supervisor-subordinate guanxi and pro-environmental behaviour. We also find that subjective norms contribute to pro-environmental behaviour while environmental concern seems to receive less consideration from employees in a Chinese society. Hence, pluralistic ignorance can also be positive if employees’ predisposition to improve supervisor-subordinate guanxi ultimately modifies the course of action.…” (see page 13).

Round 3
Reviewer 1 Report
The proposed suggestions have been made.